# Synthesis of Bimetallic BiPO_4_/ZnO Nanocomposite: Enhanced Photocatalytic Dye Degradation and Antibacterial Applications

**DOI:** 10.3390/ijms24031947

**Published:** 2023-01-18

**Authors:** Muthukumar Krishnan, Harinee Subramanian, Sathish Kumar Ramachandran, Arulmozhi Muthukumarasamy, Dineshram Ramadoss, Ashok Mahalingam, Arthur James Rathinam, Hans-Uwe Dahms, Jiang-Shiou Hwang

**Affiliations:** 1Department of Petrochemical Technology, Bharathidasan Institute of Technology Campus, University College of Engineering, Anna University, Tiruchirappalli 620 024, Tamil Nadu, India; 2Department of Physics, National Institute of Technology (NIT), Tiruchirappalli 620 015, Tamil Nadu, India; 3Department of Biomaterials, Saveetha Institute of Medical and Technical Sciences, Saveetha Dental College & Hospitals, Chennai 600 077, Tamil Nadu, India; 4Biological Oceanography Division, CSIR-National Institute of Oceanography, Dona Paula 403 004, Goa, India; 5Department of Marine Science, Bharathidasan University, Tiruchirappalli 620 024, Tamil Nadu, India; 6Department of Biomedical Science and Environmental Biology, Kaohsiung Medical University (KMU), No. 100, Shin-Chuan 1st Road, Kaohsiung 80708, Taiwan; 7Department of Marine Biotechnology and Resources, National Sun Yat-sen University, Kaohsiung 80424, Taiwan; 8Research Center for Precision Environmental Medicine, KMU—Kaohsiung Medical University, Kaohsiung 80708, Taiwan; 9Institute of Marine Biology, National Taiwan Ocean University, Keelung 20224, Taiwan; 10Center of Excellence for Ocean Engineering, National Taiwan Ocean University, Keelung 20224, Taiwan; 11Center of Excellence for the Oceans, National Taiwan Ocean University, Keelung 20224, Taiwan

**Keywords:** hydrothermal synthesis, BiPO_4_/ZnO nanocomposite, multidrug-resistant strains, photocatalyst, antibacterial efficacies, UV light irradiation

## Abstract

Multidrug-resistant strains (MDRs) are becoming a major concern in a variety of settings, including water treatment and the medical industry. Well-dispersed catalysts such as BiPO_4_, ZnO nanoparticles (NPs), and different ratios of BiPO_4_/ZnO nanocomposites (NCs) were synthesized through hydrothermal treatments. The morphological behavior of the prepared catalysts was characterized using XRD, Raman spectra, PL, UV–Vis diffuse reflectance spectroscopy (UV-DRS), SEM, EDX, and Fe-SEM. MDRs were isolated and identified by the 16s rDNA technique as belonging to *B. flexus*, *B. filamentosus*, *P. stutzeri*, and *A. baumannii*. The antibacterial activity against MDRs and the photocatalytic methylene blue (MB) dye degradation activity of the synthesized NPs and NCs were studied. The results demonstrate that the prepared BiPO_4_/ZnO-NCs (B1Z4-75:300; NCs-4) caused a maximum growth inhibition of 20 mm against *A. baumannii* and a minimum growth inhibition of 12 mm against *B. filamentosus* at 80 μg mL^−1^ concentrations of the NPs and NCs. Thus, NCs-4 might be a suitable alternative to further explore and develop as an antibacterial agent. The obtained results statistically justified the data (*p* ≤ 0.05) via one-way analysis of variance (ANOVA). According to the results of the antibacterial and photocatalytic study, we selected the best bimetallic NCs-4 for the photoexcited antibacterial effect of MDRs, including Gram ve^+^ and Gram ve^−^ strains, via UV light irradiation. The flower-like NCs-4 composites showed more effectiveness than those of BiPO_4_, ZnO, and other ratios of NCs. The results encourage the development of flower-like NCs-4 to enhance the photocatalytic antibacterial technique for water purification.

## 1. Introduction

Over the past several decades, water pollution, usually caused by dye-based industries, has often resulted in the environment posing a potential risk to human health and ecological systems [1,2,3]. Water quality is an emerging property of a complex system composed of microbial populations, and wastewater containing harmful dyes poses a major problem to the aquatic environment [4,5]. Approximately 1.84 billion metric tonnes of wastewater from industrial dyes (textiles, paper, food, and pharmaceuticals) are produced every year [6], and 17 million people die worldwide from infectious diseases caused by microbial pathogens [7]. Effluent can be purified via physical, chemical, and biological treatment technologies which are necessary to protect the aquatic environment [8]. The photocatalytic degradation in environmental remediation has attracted increasing interest owing to its eco-friendly merits and decolorization of dye-contaminated wastewater [9].

In addition, the naturally occurring microbial contamination of the aquatic environment by multidrug-resistant (MDR) microbes seriously threatens public health and national priorities around the world [10,11]. The World Health Organization (WHO) has professed that water contaminated by various types of organisms such as bacteria, viruses, and protists has become a great threat to human health [12]. Recently, microbial contamination of water bodies has had a strong impact on animal and human life. Nanotechnology has emerged through the integration of photocatalytic water treatment to develop environmentally friendly technologies for the fabrication of nanomaterials. World scientists and researchers have focused on semiconductor nanoparticles (NPs) or nanocomposites (NCs) as novel catalysts, with considerable interest in their potential applications [13]. They represent promising alternatives to photocatalytic technology that are inexpensive and highly efficient for water purification.

The water pollution crisis is a major problem that can be solved with the emergence of semiconductor photocatalysis. NPs and NCs are intensively explored because of their numerous uses in nanoelectronics, optoelectronics, nanosensors, bio-labeling, catalysis, nanomedicine, and as antibacterial agents [14]. In recent years, numerous NP and NC semiconductor photocatalysts such as ZnO [15], TiO_2_ [16], Bi_2_WO_6_ [17], Ag-doped ZnO [18], BiVO_4_/Bi_2_S_3_ [19], BiVO_4_/Bi_2_O_3_ [20], BiVO_4_/WO_3_ [21], Bi_2_O_3_ [22], and BiVO_4_/BiOI [23] have raised public interest because of their widespread applications. Among several nano-based materials, bismuth (III) phosphate (BiPO_4_) with relatively narrow band-gap energy (2.4 eV) and zinc oxide (ZnO) nanoparticles with wide band-gap (3.37 eV) and extraordinary excitation (60 meV) energy were applied in the most promising catalytic fields [24]. BiPO_4_ and ZnO have been regarded as suitable choices because their raw materials are abundant, they have a wide range of applications, and they have lower costs compared to non-metals. The synthesized nanocomposites are structurally characterized here and observed by XRD, Raman spectroscopy, PL spectroscopy, UV–Vis diffuse reflectance spectroscopy (UV-DRS), SEM, FE-SEM, and EDX. The research results indicate that the bimetallic BiPO_4_/ZnO [B1Z4-75:300; (NCs-4)] nano-flower showed much higher photocatalytic methylene blue (MB) dye degradation activity compared to separate BiPO_4_ and ZnO treatments and different ratios of BiPO_4_/ZnO composites.

## 2. Results and Discussion

### 2.1. Characterization of As-Prepared NPs and NCs Catalysts

In recent years, nanotechnology has demonstrated considerable potential applications, such as in photocatalysis, photography, optics, electronics, optoelectronics, information storage, luminescence tagging, labeling, administration of medical drugs, and imaging by employing metal particles of the nanoscale size range [25,26]. Semiconductor nano-based photocatalytic technology has gained growing attention due to its highly efficient, low-cost, and eco-friendly removal of organic dye contaminations. Crystalline properties of the as-prepared BiPO_4_-NPs (NPs-1), ZnO-NPs (NPs-2), BiPO_4_/ZnO-NCs (B1Z1-300:300; NCs-1), BiPO_4_/ZnO-NCs (B2Z1-300:150; NCs-2), BiPO_4_/ZnO-NCs (B4Z1-300:75; NCs-3), and BiPO_4_/ZnO-NCs (B1Z4-75:300; NCs-4) were first evaluated by XRD, and the results are demonstrated in Figure 1a.

The XRD results revealed the as-prepared samples’ crystalline nature, which agrees with the standard data (JCPDS No. 15-0767) for hexagonal BiPO_4_ with P2_1_/n space group symmetry. Chengsi et al. [27] reported that upon heating, BiPO_4_ undergoes a phase shift from hexagonal to monoclinic. The coordination number around Bi^3+^ differs between the hexagonal and monoclinic phases of BiPO_4_. The hexagonal BiPO_4_ is a new type of inorganic non-metal salt of oxy-acid photocatalyst, Bi^3+^ ions that are surrounded by eight near-neighbor oxygen atoms forming square anti-prism geometry around Bi^3+^, whereas in monoclinic BiPO_4_, Bi^3+^ has a coordination number of nine and it exhibits better photocatalytic activity [28]. Furthermore, the crystalline size of the as-prepared pure NPs and NCs was estimated using the Debye Scherer equation [29,30] as follows:Crystalline size D = (Kλ/βcosθ)(1)
where K is the Scherer constant (k = 0.9), λ is the wavelength of the X-ray (λ = 1.5406), β is the full width at half maximum (FWHM), and cos θ is the Bragg angle. Similarly, the diffraction peaks of ZnO are strongly correlated with the hexagonal phase ZnO reported in the JCPDS data (No. 36-1451). XRD spectra indicate that the sample ZnO nanoparticles consist of a pure phase, with no distinctive peaks for other impurities. The comparison intensities of the peaks at 21.4°, 27.2°, and 31.2° were noticeably elevated in the BiPO_4_/ZnO nanocomposites when the concentration of BiPO_4_ increased. This indicates that no other impurities were found, demonstrating the purity of BiPO_4_/ZnO nanocomposites such as B1Z1 (300:300), B2Z1 (300:150), B4Z1 (300:75), and B1Z4 (75:300). The BiPO_4_/ZnO composite showed that both the BiPO_4_ phase and the cubic ZnO phase were co-existing, demonstrating that a mixture of BiPO_4_ and ZnO is the predominant form of BiPO_4_/ZnO nanocomposites [31].

The Raman spectra for as-synthesized nanocomposites are shown in Figure 1b. The observed intense bands at lower wavenumber regions at 168 cm^−1^, 230 cm^−1^, and 280 cm^−1^ can be assigned to the stretching vibration of Bi–O bonds. The two bands centered at higher energies of 1040 cm^−1^ and 966 cm^−1^ are attributed to the asymmetric (γ3) and symmetric (γ1) stretching vibrations of the P–O bonds in the PO_4_ group, respectively. The bands in the region at 598 cm^−1^ and 555 cm^−1^ correspond to the γ4 bending vibration modes of the PO_4_ tetrahedron. The weak bands at 460 cm^−1^ and 404 cm^−1^ can be ascribed to the γ2 bending modes of the PO_4_ units. A sharp, strong, and dominant peak at 438 cm^−1^ is attributed to the non-polar, high-frequency optical phonon Raman mode (E_2H_), a characteristic peak of the wurtzite hexagonal phase of ZnO. In addition, the peak at 300 cm^−1^ is assigned to the E_2H_-E_2L_ (multi-phonon scattering) mode and the A1 (longitudinal optical) mode arising at 1160 cm, respectively. A peak that appears at 935 cm^−1^ corresponds to a second-order multi-phonon scattering mode (2E_2H_ + E_2L_).

Figure 1c shows the UV–Vis diffuse reflectance spectra (UV-DRS) of the as-prepared BiPO_4_-ZnO composite photocatalyst. Strong absorption bands centered at 216 nm and 256 nm have been assigned to the charge transfer (C-T) transition, which originates predominantly from the hybrid electrons of Bi^3+^ and O^2−^. The light absorbance of pure ZnO is in the UV region, with an absorption edge at 355 nm, and a band-gap around 3.8 eV. Due to the electron transitions from the valence band to the conduction band (O_2p_-Zn_3d_), a band at 360 nm might be attributed to the intrinsic band-gap absorption of ZnO. The corresponding band-gap energies (E_g_) of composites were calculated using K-M theory and plotted as [*F(R) hν*]^2^ versus photon energy, and it was found to be 3.42 eV, which is shown in the inset of the image. It seems pertinent that the as-synthesized nanocomposite can be excited under UV light exposure, consequently resulting in higher photocatalytic degradation activity.

The PL spectra of the pure BiPO_4_, ZnO, and combined BiPO_4_/ZnO nanocomposites are presented in Figure 1d. The PL excitation spectrum was measured by monitoring the emission wavelength of 325 nm and the visible region of broad peaks from 325 to 600 nm [32]. The Bi^3+^ exhibits an emission band centered at 356 nm under UV light excitation due to the transition from ^3^P_1_ to ^1^S_0_ of Bi^3+^. A sharp intensity peak observed in the ultraviolet (UV) region around 395 nm corresponds to near-band-edge excitation emission of ZnO. A blue-green emission is present in the range of 470–575 nm. The PL emission peaks at lower energy correspond to zinc vacancies (V_zn_) and antisite defects (O_zn_). Around 575 nm, green band emissions are ascribed to the existence of a single ionized oxygen vacancy in ZnO. The emission is caused by the radioactive recombination of a photogenerated hole with an electron occupying the oxygen vacancy. It is known that O vacancy is one of the most important factors for narrowing the band-gap of ZnO. Visible luminescence is mainly due to defects related to deep-level emissions, such as Zn interstitials and oxygen vacancies [33].

### 2.2. In Vitro Antibacterial Susceptibility Testing

The antibacterial efficacy against 16S rDNA-recognized DRS (*n* = 4), including Gram ve^+^ and Gram ve^−^ strains, were assessed through NP and NC catalysts (NPs-1, NPs-2, NCs-1, NCs-2, NCs-3, and NCs-4). The experimental results exhibited the as-prepared BiPO_4_/ZnO-NCs (B1Z4-75:300; NCs-4) nanocomposites displaying greater antibacterial activities compared to pure BiPO_4_-NPs (NPs-1), ZnO-NPs (NPs-2), BiPO_4_/ZnO-NCs (B1Z1-300:300; NCs-1), BiPO_4_/ZnO-NCs (B2Z1-300:150; NCs-2), and BiPO_4_/ZnO-NCs (B4Z1-300:75; NCs-3). The results demonstrate that the as-prepared BiPO_4_/ZnO-NCs (B1Z4-75:300; NCs-4) caused a maximum growth inhibition of 20 mm against the MDR bacteria *A. baumannii* and a minimum growth inhibition of 12 mm against *B. filamentosus* at 80 μg mL^−1^ concentrations of the NPs and NCs, as displayed in Figure 2. The antibacterial results revealed a greater sensitivity at lower doses, indicating that concentration dependency is crucial. Gram ve^+^ bacteria have multi-layered peptidoglycan cell walls that are substantially thicker than those of Gram-negative bacteria, with a complex cell wall structure consisting of a thin layer of peptidoglycan between the outer plasma and the cytoplasmic membrane. This result clearly demonstrates that the 80 μg mL^−1^ concentrations of as-prepared BiPO_4_/ZnO-NC catalysts (B1Z4-75:300; NCs-4) affect MDRs directly, leading to cell death [34]. Antibacterial activities were demonstrated by analysis of variance (one-way ANOVA) for differences between means, demonstrating the statistical significance (*p* ≤ 0.05) of the data, shown in Table 1. The antibacterial activity of as-prepared pure nanoparticles and nanocomposites (NPs-1, NPs-2, NCs-1, NCs-2, NCs-3, and NC-4) at four different concentrations (20, 40, 60, and 80 μg mL^−1^) showed the following values for sum of squares, mean square, F-value, and probability > F* of MDRs: 4.5, 2.25, 0.11947, and 0.88878 for NPs-1; 8, 4, 1.09091, and 0.37652 for NPs-2; 8, 4, 1.09091, and 0.37652 for NCs-1; 8, 4, 1.5, and 0.27402 for NCs-2; 8, 4, 0.6, and 0.56936 for NCs-3; and 8, 4, 0.34286 and 0.71862 for NC-4, respectively. The results show that there is insufficient evidence to statistically express the significance of antibacterial activities at the four different concentrations.

### 2.3. SEM and Fe-SEM Characterization of NCs-4 Catalysts

The morphological shape and crystallinity size of as-prepared NCs-4 were characterized by SEM and Fe-SEM analysis based on its strong antibacterial activity towards MDRs. The representative SEM images of hexagonal and monoclinic BiPO_4_ and ZnO showed nanoparticles with mostly spherical and crystalline and polydispersed-like morphology, as shown in Figure 3a. Fe-SEM images demonstrate that most of the particles are flower-shaped, and the particle size distribution chart shows that the prepared nanocomposites have a size range from 30 to 60 nm with an average diameter of about 45 nm, shown in Figure 3b. In the case of BiPO_4_/ZnO, the flower shape was not observed; however, the Fe-SEM image showed the formation of multifaceted clusters of flower morphologies. Because ZnO^+^ has a substantially lower atomic radius than Bi^3+^, doping of BiPO_4_ with ZnO resulted in a crystallographic strain of the BiPO_4_ lattice. These defect sites in the partially distorted BiPO_4_ lattice served as additional nucleation centers. Simultaneous development of numerous nuclei delayed the lattice’s pseudo-one-dimensional growth and resulted in multidimensional clusters with reduced overall length. The obtained nanocomposite size distribution for BiPO_4_/ZnO using SEM also agrees well with the sizes provided by Fe-SEM. The EDS elemental mapping results are shown in Figure 3c. The elements Zn, Bi, P, and O present in the nanocomposite spectrum seem to belong to BiPO_4_/ZnO-NCs.

### 2.4. Photocatalytic Experiments

The photocatalytic performances of as-prepared NCs-4 nanocomposites were investigated by degrading methylene blue (MB) dyes under UV light irradiation. Figure 4a shows the photocatalytic degradation activity of the composites under UV light irradiation. Among all samples, BiPO_4_/ZnO-NCs (B1Z4-75:300; NCs-4) show the highest degradation efficiency under UV light, and almost 96% of the dye (MB) was degraded within 60 min of irradiation, while BiPO_4_/ZnO-NCs (B1Z4-300:300; NCs-1) reached a constant value after 90 minutes of irradiation and caused 84% degradation. All the pure BiPO_4_-NPs showed low photocatalytic activity when compared to NPs and NCs. A pseudo-first-order kinetic model was employed to calculate the rate constant [35]. The plot of ln C_t_/C_0_ versus time follows a linear fit, showing that the photocatalytic degradation reaction of MB dye follows pseudo-first-order kinetics, and the corresponding rate constant (*k*) values were calculated (Figure 4b). After four cycles of the composite, there was no obvious decrease in the photocatalytic MB dye degradation activity, as shown in Figure 4c. Reusing the NCs-4 showed no obvious change in the stability, crystal morphology, or degradability under UV light exposure. The photocatalytic dye degradation activity of NCs-4 was examined with four cycles carried out towards the degradation of MB dye under UV light exposure.

The photocatalytic inactivation experiments (MDRs and MB dye) were studied under UV light irradiation with the NCs-4 catalyst. The enumeration study displayed [×10^7^–10^8^ CFUs mL^−1^] initial counts of MDR colonies compared to photocatalytically inactivated suspension plates at time intervals of 5 to 20 minutes. The NCs with UV light treatment demonstrated considerable growth inhibition of MDR colonies after 20 minutes, as indicated in Table 2. The antibacterial interaction of BiPO_4_/ZnO nanocomposites directly affected the identified osmotic shock of photo-induced holes (H^+^) and the hydroxyl radicals (^-^OH) that directly attacked MDR cell walls. The nano-flower NCs-4 bimetallic nanocomposites were evidenced as an existent photocatalytic agent that was utilized for MB dye degradation in addition to antibacterial effects [36]. Moreover, in order to assess the activity of the photocatalyst results, they were compared with those of ZnO, TiO_2_, Fe_2_O_3_, BiVO_4_, nano-heterojunction AgFeO_2_/ZnO, ZnO-based heterogeneous, Ag/ZnO heterostructural, polyaniline/CdO nanocomposite, Bi_2_WO_6_/BiVO_4_, and Bi_2_WO_6_/BiOI p-n heterojunction, which have been widely used for the degradation of organic dyes and controlling microbes [37,38,39,40,41,42,43,44,45,46]. Compared to the standard of AEROXIDE^®^ TiO2 P-25, our resultant nanopowders are also waterborne and have adequate solubility in water, with a solid white form as well as free-flowing powder with a pH value of 4.5. In addition, similar to AEROXIDE^®^ TiO_2_ P-25, our resultant bimetallic BiPO_4_/ZnO nanocomposites have a high specific surface area. Due to its unique flower crystalline structure, it is suitable for many catalytic and especially photocatalytic applications. This result clearly shows that the produced nanostructures have equal potential as the standard AEROXIDE^®^ TiO_2_ P-25 and can be applicable to photocatalytic degradation, especially on organic pollutants. The results demonstrate that the BiPO_4_/ZnO-NCs (B1Z4-75:300; NCs-4) catalyst showed significantly superior photocatalytic activity under UV light sources. The photocatalytic degradation of organic pollutants is crucial, as it effectively disintegrates the toxicants that pollute environmental waters such as surface and groundwater, causing significant pollution.

Initial count—The test culture suspensions were prepared in sterile 0.85% saline matching an optical density of 0.5 McFarland standards corresponding to 10^7^–10^8^ (CFUs mL^−1^).Gram ve^+^ (n = 2)—*Bacillus flexus*—NCBI number: MN045189 (*B. flexus*), *Bacillus filamentosus*—NCBI number: MN045186 (*B. filamentosus*).Gram ve^−^ (n = 2)—*Pseudomonas stutzeri*—NCBI number: MN045185 (*P. stutzeri*), and *Acinetobacter baumannii*—NCBI number: MN045188 (*A. baumannii*).

## 3. Experimental Section

### 3.1. Materials and Reagents

Commercially available analytical-grade (AG) chemicals were used as precursors without purification. Purified bismuth (III) nitrate pentahydrate [Bi(NO_3_)_3_.5H_2_O], disodium hydrogen phosphate anhydrous (Na_2_HPO_4_), potassium hydroxide (KOH), and zinc acetate dihydrate [Zn(CH_3_COO)_2_.2H_2_O] (99.99%) were procured from Sigma-Aldrich, India. The microbial culture media, standard disks, sterile swabs, and Hi-Antibiotic Zone Scale-C were purchased from Hi-Media (Mumbai, India), and all aqueous solutions and chemicals were prepared using Milli-Q water (Millipore, Burlington, MA, USA).

### 3.2. Collection of Microbial Cultures

Multidrug-resistant strains (MDRs) were isolated from Tamil Nadu governmental hospital wastewater at Srirangam (latitude 78°68′ east and longitude 10°87′ north) and Tiruchirappalli (latitude 78°40′ east and longitude 10°48′ north), India. Test organisms, which were 16S rDNA-confirmed MDRs, were used for antibacterial and photocatalytic inactivation studies. The MDRs belonged to Gram ve^+^ (n = 2) and Gram ve^−^ (n = 2) strains such as *Bacillus flexus*—NCBI number: MN045189 (*B. flexus*); *Bacillus filamentosus*—NCBI number: MN045186 (*B. filamentosus*); *Pseudomonas stutzeri*—NCBI number: MN045185 (*P. stutzeri*); and *Acinetobacter baumannii*—NCBI number: MN045188 (*A. baumannii*). A phylogenetic tree was constructed by MEGA software version 4.1, shown in Figure 5.

### 3.3. Preparation of Catalysts

#### 3.3.1. Synthesis of BiPO_4_ Nanoparticles (NPs)

Precursor solutions of Bi(NO_3_)_3_.5H_2_O, 4 mmol were dissolved in 60 mL of Milli-Q water along with 5 mL of concentrated HNO_3_ under constant stirring until the precursor dissolved. Subsequently, 4 mmol of Na_2_HPO_4_ was added drop-wise. Then, the mixed solvent was stirred for 3 h at 70 °C to form a well-homogenized white suspension. Finally, the obtained suspension was transferred to a 100 mL Teflon-lined stainless steel sealer, then autoclaved and maintained at 200 °C for 15 h. After autoclaving, it was allowed to cool for 15 h at room temperature (28 ± 2 °C). After that, the collected solution was centrifuged at 6000 rpm for 30 minutes. Consequently, the supernatant was washed several times with Milli-Q water and dried at 60 °C under vacuum, and the final product containing BiPO_4_ was ground into a powder.

#### 3.3.2. The Synthesis of Zinc Oxide Nanoparticles (ZnO-NPs)

Wet chemical method ZnO nanoparticles were prepared using a 250 mL round bottom flask (RBF). In a typical process, the aqueous solution of polyethylene glycol (PEG) and 1 M zinc acetate dehydrate [Zn(CH_3_COO)_2_.2H_2_O] were added separately to the flask, and each was dissolved through ultrasound sonication in 30 mL of Milli-Q water for several minutes. Thereafter, 25 mL of the 0.2 M KOH solution was added drop-wise along the walls of the RBF, forming a transparent white solution, and placed on a magnetic stirrer at a constant temperature of 90 °C for 3 h. These solutions were reacted to produce ZnO precipitates that formed a white suspension that was cooled to room temperature. After precipitation, the solution was centrifuged for 30 minutes at 3000 rpm. The supernatant was washed several times using Milli-Q water and dried in a vacuum at 60 °C for 24 h, resulting in the formation of ZnO nano-powder. Finally, it was ground in a mortar to powder and stored in the refrigerator at 4 °C until further studies. This experiment was performed eight times at different concentrations under the same process conditions.

#### 3.3.3. Preparation of Bimetallic BiPO_4_/ZnO Nanocomposite (NCs)

Note that the above process is favorable for the subsequent homogeneous deposition of BiPO_4_/ZnO nanocomposites. Pure BiPO_4_-NPs (NPs-1) and ZnO-NPs (NPs-2) and the different molar ratios of BiPO_4_/ZnO composites such as BiPO_4_/ZnO-NCs (B1Z1-300:300; NCs-1), BiPO_4_/ZnO-NCs (B2Z1-300:150; NCs-2), BiPO_4_/ZnO-NCs (B4Z1-300:75; NCs-3), and BiPO_4_/ZnO-NCs (B1Z4-75:300; NCs-4) were prepared and then stored in a refrigerator at 4 °C until further studies.

### 3.4. Characterization and Property Measurements of As-Prepared NPs and NCs

Crystallite size and phase purity of as-prepared pure nanoparticles and nanocomposites (NPs-1, NPs-2, NCs-1, NCs-2, NCs-3, and NC-4) were analyzed by XRD. The X-ray diffraction (XRD) pattern, recorded by using the Rigaku ULTIMA III with Cu-*k_α_* anode radiation (*λ* = 1.54056 Å), was operated at 40 kV over a 2*θ* collection range of 20–80° with a step size of 0.02° and a scan rate of 4° per minutes. Raman spectra were recorded for all samples equipped with the laser source at the wavelength (*λ)* 514.5 nm recorded at room temperature. Photoluminescence (PL) spectra were employed at room temperature with 325 nm as the excitation wavelength, and He-Cd laser as the source of excitation. The optical properties of the composites were analyzed by UV–Vis diffuse reflectance spectroscopy (UV-DRS; JASCO; UV—1700) at a wavelength range of 200–800 nm. The surface morphology, chemical composition, crystallite size, and structure of nanoparticles were characterized by a scanning electron microscope (SEM) (HITACHI; S-3000H) equipped with energy-dispersive spectroscopy (EDS). The size distribution and average size of the nanoparticles were estimated on a field emission scanning electron microscope (Fe-SEM; JSM-6360LA).

### 3.5. In Vitro Antibacterial Susceptibility Testing

The antibacterial efficacies of the pure BiPO_4_-NPs (NPs-1), ZnO-NPs (NPs-2), BiPO_4_/ZnO-NCs (B1Z1-300:300; NCs-1), BiPO_4_/ZnO-NCs (B2Z1-300:150; NCs-2), BiPO_4_/ZnO-NCs (B4Z1-300:75; NCs-3), and BiPO_4_/ZnO-NCs (B1Z4-75:300; NCs-4) were tested against 16S rDNA-confirmed MDRs by the well diffusion technique. The MDRs (n = 4), including Gram ve^+^ and Gram ve^−^ strains meeting McFarland standards corresponding to 10^8^ CFUs mL^−1^, were swabbed on Petri dishes with Muller Hinton agar (MHA) medium. The NPs and NCs at four different concentrations (20, 40, 60, and 80 μg mL^−1^) of pure nanoparticles and nanocomposites (NPs-1, NPs-2, NCs-1, NCs-2, NCs-3, and NC-4) were loaded (6 mm diameter well) in each Petri dish. Then, the loaded Petri dishes were incubated at 37 ± 2 °C for 24–48 h. After incubation, the growth inhibition was measured by using the Hi-Antibiotic zone measuring Scale-C [47,48].

### 3.6. Statistical Analysis

The antibacterial susceptibility results are expressed as the mean + SD of the inhibition zone (mm) of three replicates. To understand the relationship between the variables, we used one-way analysis of variance (ANOVA) with ORIGIN8.0 version software; the results were statistically significant if *p* < 0.05.

### 3.7. Photocatalytic Activity Test

#### 3.7.1. Photocatalytic Efficacies of NPs and NCs Catalyst

The photocatalytic dye degradation efficiency was performed under UV light exposure in the presence of NP and NC catalysts (NPs-1, NPs-2, NCs-1, NCs-2, NCs-3, and NC-4). Each NP and NC catalyst was dispersed in 200 mL of a 10^−5^ M aqueous MB dye solution. The NP and NC catalysts and multidrug-resistant strain (MDR) suspension were mixed prior to light irradiation. Before UV light irradiation, the suspension (dye and bacterial strains) was magnetically stirred at 150 rpm constantly for 20–30 minutes in dark conditions to obtain an adsorption/desorption equilibrium in the presence of the catalyst. After that, the suspensions were irradiated by UV light (Philips-UV light (150 W); λ = 365 nm). During the test, the degradation efficacy was tested at appropriate time intervals; then, 2 mL of suspension was separated and centrifuged at 10,000 rpm for 30 minutes. The degradation efficiency of MB dye concentrations was monitored through a UV–visible spectroscope (Model–SHIMADZU 1700; 400–800 nm). The dye degradation efficiency percentage (%) was calculated according to the equation by Dai et al. [49]:Degradation % = (*C*_0_ − *C*/*C*_0_) × 100(2)
where *C*_0_ is the initial concentration of MB dye (mg L^−1^), and *C* is the remaining MB dye concentration (mg L^−1^) of the aqueous solution at a given time under UV light irradiation.

Photocatalytic MB dye degradation follows pseudo-first-order kinetics, and the rate constant (k) was estimated by Equation (3):−ln (*C_t/_C*_0_) = *kt*(3)
where *C*_t_ is the concentration of MB after irradiation, *C*_0_ is the concentration of dye before UV light irradiation for reaction time 0 to *t* (mg L^−1^), and *k* represents the photocatalytic reaction rate (minutes^−1^).

#### 3.7.2. Photocatalytic Inactivation Experiments

Based on the antibacterial susceptibility test and photocatalytic dye degradation efficiency results, we chose BiPO_4_/ZnO-NCs (B1Z4-75:300; NCs-4) for a photocatalytic inactivation experiment. The photocatalytic inactivation effects on MDR colonies were investigated through an enumeration test, shown in Figure 6. The NCs-4 catalyst was dispersed in 200 mL of 10^−5^ M MB dye solution. Then, 100 μL of each MDR (n = 4), including Gram ve^+^ and Gram ve^−^ strains, was blended prior to UV light exposure. Prior to UV light exposure, the MDRs and MB dye suspension were stirred at 150 rpm for 30 minutes in the dark to achieve an adsorption/desorption equilibrium. Then, the suspensions were irradiated using UV light.

For the experiment, 5 mL of the treated (MDRs and MB dye) suspension was collected every 5 minutes and diluted using Milli-Q water. Then, 200 μL of the diluted treated suspension was uniformly spread on nutrient agar (NA) plates, which were incubated at 37 ± 2 °C for 24 h. The control was conducted according to the above procedure without adding the NCs-4 catalyst. The number of viable colonies was counted with a digital colony counter (make: Medica Gmp; model: 0671m) and is expressed as CFUs mL^−1^. Finally, the survival rate was calculated through the following equation:Photocatalytic inactivation (%) = (N _Survivor/_N _Control_) × 100(4)
where N _Control_ is the MDR colony initial count and N _Survivor_ is the photocatalytic inactivation-treated MDR colony count.

### 3.8. Photocatalytic Cycle Test

The reusability/stability of the NCs-4 catalyst was tested for four cycles under UV light exposure for dye degradation. At the end of the experiment, the NCs-4 catalyst was separated by centrifuging it at 10,000 rpm for 20 minutes and discarding the supernatant solution. Subsequently, the obtained NCs-4 catalyst pellet was rinsed several times with Milli-Q water. Finally, the obtained NCs-4 catalyst pellet was dried at 60 °C for 5 h in a hot air oven; the obtained catalyst powder was further reused for successive degradation. The same procedure was followed for all repeated tests [50].

## 4. Conclusions

Based on the above analysis, an efficient hydrothermal method for the synthesis of BiPO_4_/ZnO nanocomposites with different molar ratios was developed. From XRD analysis, the structure of BiPO_4_ and ZnO was confirmed. Nanocomposite size was found in the range of 30–60 nm. The charge transfer transition was observed in UV-DRS. The emission and excitation spectra were observed, and the sample showed green emission. The antibacterial activity of flower-like BiPO_4_/ZnO-NCs (B1Z4-75:300; NCs-4) provided a maximum growth inhibition of 18 mm against *P. stutzeri* and a minimum growth inhibition of 12 mm against *B. filamentosus* at a concentration of 80 μg mL^−1^. The photocatalytic inactivation of MDRs, including Gram ve^+^ and Gram ve^−^ strains, via UV light irradiation using as-prepared NCs-4 exhibited maximal antibacterial activity. The photocatalytic studies on decolorization of MB dye were conducted using all composites (NPs and NCs) that have been established under UV light irradiation. NCs-4 bimetallic nanocomposites exhibited sufficiently enhanced activity. The flower-like NCs-4 bimetallic nanocomposites exhibited the highest degradation of methylene blue (MB) dye of 96% in 60 minutes under UV light irradiation, which supports environmentally safe and cost-effective water treatment.

## Figures and Tables

**Figure 1 ijms-24-01947-f001:**
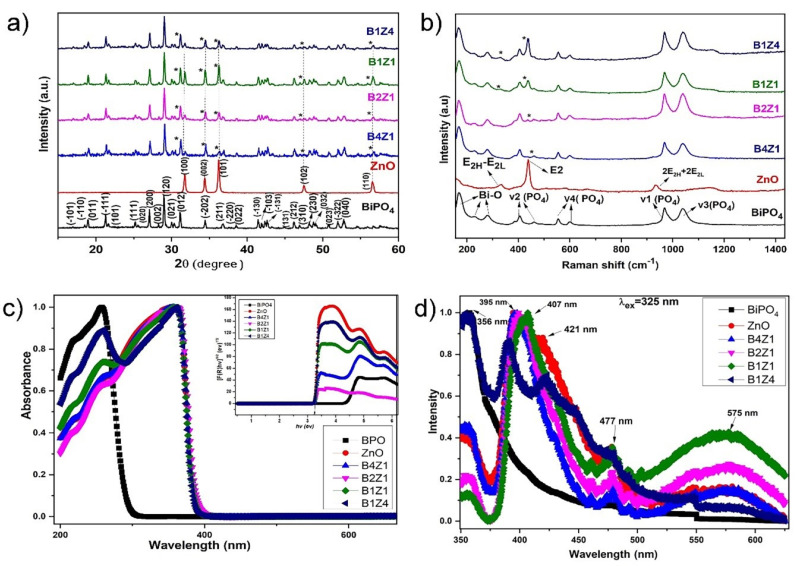
Characterization of synthesized NPs and different concentrations of BiPO_4_ and ZnO loadings of NCs: (**a**) X-ray diffraction pattern *; (**b**) Raman spectra; (**c**) UV–vis spectroscopy absorbance spectra; (**d**) photoluminescence (PL) spectra.

**Figure 2 ijms-24-01947-f002:**
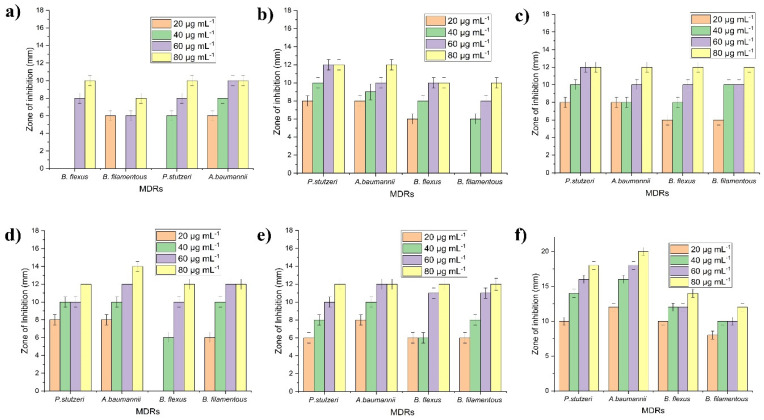
In vitro antibacterial efficacy against 16S rDNA-identified MDRs of NPs and NCs with different concentrations (20, 40, 60, and 80 μg mL^−1^) by the well diffusion method: (**a**) pure BiPO_4_-NPs (NPs-1); (**b**) ZnO-NPs (NPs-2); (**c**) BiPO_4_/ZnO-NCs (B1Z1-300:300; NCs-1); (**d**) BiPO_4_/ZnO-NCs (B2Z1-300:150; NCs-2); (**e**) BiPO_4_/ZnO-NCs (B4Z1-300:75; NCs-3); (**f**) BiPO_4_/ZnO-NCs (B1Z4-75:300; NCs-4).

**Figure 3 ijms-24-01947-f003:**
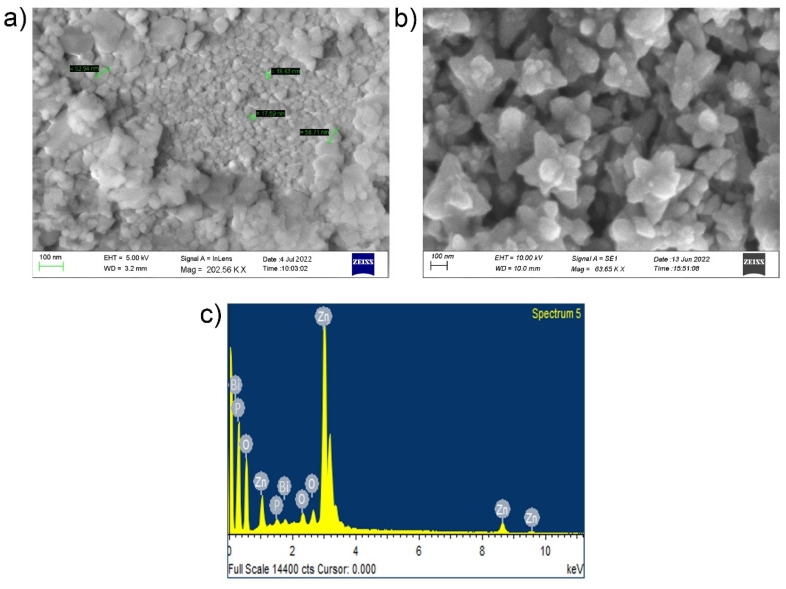
(**a**) SEM observation; (**b**) Fe-SEM observation; (**c**) EDS spectrum image of flower-like BiPO_4_/ZnO-NCs (B1Z4-75:300; NCs-4).

**Figure 4 ijms-24-01947-f004:**
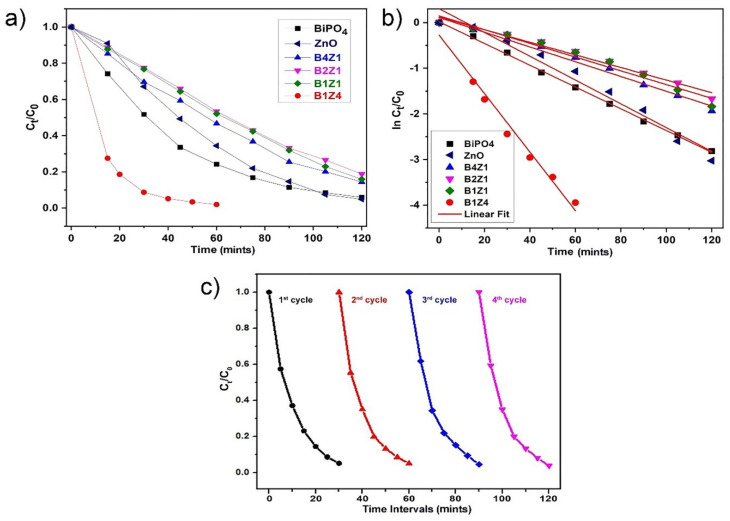
(**a**) Photocatalytic degradation of NPs and different concentrations of BiPO_4_ and ZnO loadings NCs under UV light exposure. (**b**) Pseudo-first-order-kinetics plot for the catalytic degradation of MB dye. (**c**) Photocatalytic reusability/stability test after repetition reaction of flower-like BiPO_4_/ZnO-NCs (B1Z4-75:300; NCs-4).

**Figure 5 ijms-24-01947-f005:**
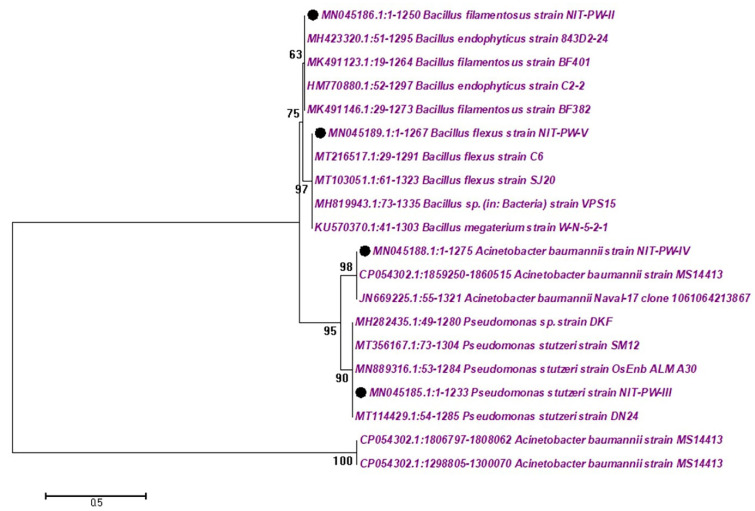
The isolated multidrug-resistant strains (MDRs) constructed by the phylogenetic tree are based on about 1200 partial bases of 16S rDNA gene sequences.

**Figure 6 ijms-24-01947-f006:**
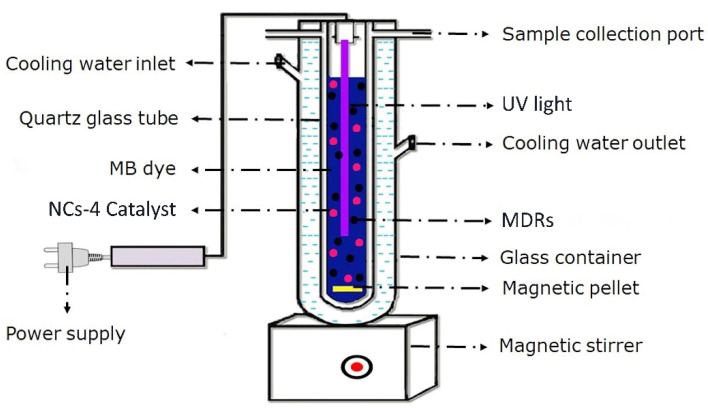
Photocatalytic inactivation test setup under UV light exposure.

**Table 1 ijms-24-01947-t001:** ANOVA in vitro antibacterial susceptibility testing performed with as-synthesized NPs and NCs at different concentrations (20, 40, 60, and 80 μg mL^−1^).

Concentration	DF	Sum of Squares	Mean Square	F-Value	Probability > F *
BiPO_4_-NPs (NPs-1)					
Model	2	4.5	2.25	0.11947	0.88878
Error	9	169.5	18.83333		
Total	11	174			
ZnO-NPs (NPs-2)					
Model	2	8	4	1.09091	0.37652
Error	9	33	3.66667		
Total	11	41			
BiPO_4_/ZnO-NCs (B1Z1-300:300; NCs-1)					
Model	2	8	4	1.09091	0.37652
Error	9	33	3.66667		
Total	11	41			
BiPO_4_/ZnO-NCs (B2Z1-300:150; NCs-2)					
Model	2	8	4	1.5	0.27402
Error	9	24	2.66667		
Total	11	32			
BiPO_4_/ZnO-NCs (B4Z1-300:75; NCs-3)					
Model	2	8	4	0.6	0.56936
Error	9	60	6.66667		
Total	11	68			
BiPO_4_/ZnO-NCs (B1Z4-75:300; NCs-4)					
Model	2	8	4	0.34286	0.71862
Error	9	105	11.66667		
Total	11	113			

* *p* < 0.001 indicates that the variable means are significantly different.

**Table 2 ijms-24-01947-t002:** Photocatalytic inactivation test by using bimetallic nanocomposites BiPO_4_/ZnO-NCs (B1Z4-75:300; NCs-4) catalyst under UV light exposure.

Test Organisms	Photocatalytic Inactivation Experiments				
	Catalyst NCs-4 and UV Light Irradiation at Different Time Intervals (CFUs mL^−1^)				Initial Count (Before Treatment)
	After 5 min	After 10 min	After 15 min	After 20 min	
Multidrug-Resistant Strains (MDRs)					
Gram ve^+^	50.4 [×10^5^]	36.0 [×10^4^]	19.2 [×10^3^]	10.2 [×10^2^]	10^7^–10^8^ (CFUs mL^−1^)
Gram ve^−^	44.2 [×10^5^]	52.4 [×10^4^]	22.8 [×10^3^]	8.6 [×10^2^]	10^7^–10^8^ (CFUs mL^−1^)

## Data Availability

Not applicable.

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
