# Peer review of "Synthesis of Bimetallic BiPO4/ZnO Nanocomposite: Enhanced Photocatalytic Dye Degradation and Antibacterial Applications"

_ijms, 2023, doi:10.3390/ijms24031947_

Round 1

Reviewer 1 Report

Manuscript entitled "Synthesis of Bimetallic BiPO4/ZnO Nanocomposite: Enhanced Photocatalytic Dye Degradation and Antibacterial Applications" contains very interesting results. However, in my opinion, the manuscript in its current form should not be published. It requires significant corrections.

In order to assess the activity of the photocatalyst, the results should be compared with a known reference system. I suggest performing an analogous photocatalytic experiment in the presence of AEROXIDE® TiO2 P-25 (EVONIC, Germany; https://www.productcenter.coating-additives.com/pdf/regulatory/rds_ftp/AEROXIDE_TIO2_P_25.pdf).

In addition, the text needs to be thoroughly redrafted. Large parts of it are chaotic (e.g. introduction) and/or unclear.

Abbreviations are not used in the abstract.

No clear explanations for Table 1.

The level of inhibition should be expressed in relative units, eg %.

What is "mints" (L. 213, 214 ect.)?

Chapter 3.3.3 lacks precise information on the preparation of the BiPO4/ZnO catalytic system.

Catalyst designations in the text are convoluted and inconsistent.

Conclusions should not be a compilation of numerical results.

Where are the results obtained using equation (4) described?

What was the purpose of the experiment using photocatalysts without irradiation (Chapters 2.2 and 3.5)?

Author Response

REVIEWER 1

Comments and Suggestions for Authors

Manuscript entitled "Synthesis of Bimetallic BiPO4/ZnO Nanocomposite: Enhanced Photocatalytic Dye Degradation and Antibacterial Applications" contains very interesting results. However, in my opinion, the manuscript in its current form should not be published. It requires significant corrections.

QUERRY

In order to assess the activity of the photocatalyst, the results should be compared with a known reference system. I suggest performing an analogous photocatalytic experiment in the presence of AEROXIDE® TiO2 P-25 (EVONIC, Germany; https://www.productcenter.coating-additives.com/pdf/regulatory/rds_ftp/AEROXIDE_TIO2_P_25.pdf). In addition, the text needs to be thoroughly redrafted. Large parts of it are chaotic (e.g. introduction) and/or unclear.

Author response: We rewrote several parts of the text, with particular focus on the INTRODUCTION.

In order to assess the activity of the photocatalyst results were compared from ZnO, TiO2, Fe2O3, BiVO4, nano-heterojunction AgFeO2/ZnO, ZnO-based heterogeneous, Ag/ZnO heterostructural, polyaniline/CdO nanocomposite, Bi2WO6/BiVO4, Bi2WO6/BiOI p-n heterojunction have been widely used for the degradation of organic dyes and controlling microbes [37-46]. Bimetallic BiPO4/ZnO nanocomposites have a very high specific surface area. Due to its unique ratio of flower crystalline structure, it is suitable for many catalytic and especially photocatalytic applications. The results demonstrated the BiPO4/ZnO-NCs (B1Z4-75:300; NCs-4) catalyst showed significantly superior photocatalytic activity under UV-light sources. The photocatalytic degradation of organic pollutants is crucial, as it effectively disintegrates the toxicant that pollutes environmental waters such as surface and groundwater, causing significant pollution. 

QUERRY

Abbreviations should not be used in the abstract.

Author response: We eradicated ABBREVIATIONS from the ABSTRACT.

QUERRY

No clear explanations for Table 1.

Author response: We provided clear explanations throughout the tables and also for Table 1. The level of inhibition should be expressed in relative units, e.g. %. We replaced inhibition levels by the relative unit “percentage”.

QUERRY

What is "mints" (L. 213, 214 etc.)?

Author response: This typo was replaced by mints, meaning minutes.

QUERRY

Chapter 3.3.3 lacks precise information on the preparation of the BiPO4/ZnO catalytic system.

Author response: We provided precise information of how the BiPO4/ZnO catalytic system was prepared. Catalyst designations in the text are convoluted and inconsistent. We arranged catalyst designations of the revised version in a consistent way.

QUERRY

Conclusions should not be a compilation of numerical results.

Author response: We revised the CONCLUSION substantially and avoided the mentioning of numerical results.

QUERRY

Where were the results described that were obtained using equation (4)?

Author response: The results described using equation (4) are shown in Table 2: Photocatalytic inactivation test by using bimetallic nanocomposites BiPO4/ZnO-NCs (B1Z4-75:300; NCs-4) catalyst under UV light exposure.

QUERRY

What was the purpose of the experiment using photocatalysts without irradiation (Chapters 2.2 and 3.5)?

Author response: Bimetallic nanocomposites BiPO4/ZnO-NCs (B1Z4-75:300; NCs-4) catalyst and Multi-drug resistant strains (MDRs) suspension was mixed prior to light irradiation. Before UV irradiation, the suspension (dye and bacterial strains) was magnetically stirred at 150 rpm constantly for 20 - 30 mints at dark conditions to obtain an adsorption/desorption equilibrium in the presence of the catalyst. After that, the suspensions were irradiated by under UV light [Philips-UV light (150 W); λ=365 nm].

Reviewer 2 Report

The manuscript presents an efficient hydrothermal method for the synthesis of BiPO4/ZnO nano-composites, which possess antibacterial effects and good photocatalytic performances for MB dye degradation. There are a few things need to be addressed:

1. The format of some references (20, 23, 27, and so on) is incorrect. The family name must be written in full and initials used to represent given (firstnames. Also need to revise the sections citing the reference in the manuscript (for example:  XXX et al). 

2. (page 4) The author mentioned Figure 3 before Figure 1 and 2 . The numbering of figures should follow the order which they were mentioned in the manuscript.  

3. (page 6) The author mentioned "The results demonstrate that as-prepared nanocomposites BiPO4/ZnO-NCs (B1Z4-75:300; NCs-4) caused maximum growth inhibition of 20 mm against the MDRs bacteria A. baumannii and a minimal of 12 mm against B. filamentosus at 80 μg mL−1 concentrations than the NPs and NCs displayed in Figure. 4." Any explanation of why NCs-4 is better than other NCs and NPs in antibacterial efficacy? Does NCs-4 show best antibacterial efficacy among all NCs and NPs at all concentrations and MDRs?

4. (page 6) In Table 1, the Probability > F* all larger than 0.05, the author should conclude explicitly that there is no sufficient evidence to say that there is a statistically significant difference in antibacterial activities among the four different concentrations. This is the purpose of ANOVA test. 

5. (Page 7) The author mentioned "Fe-SEM images demonstrate that most of the particles are flower shape" but "SEM image of hexagonal and monoclinic BiPO4 and ZnO showed nanoparticles with mostly spherical and crystalline and polydispersed-like morphology", and "The obtained nanocomposite size distribution on BiPO4/ZnO using SEM also agrees well with the sizes provided by Fe-SEM“. Can the author explain why Fe-SEM has better ability than SEM to characterize the morphology of particles?

6. In Figure 4, the author discussed the antibacterial efficacy of various NPs and NCs. Is there any SEM and Fe-SEM characterization of these NPs and NCs? Do they show nanoflower or other structures? 

7. (page 7,13, Figure 6(c) and so on) The author mentioned "XX mints". If that's XX minutes, the abbreviation should be "XX mins".

8. (page 7) The author mentioned "the prepared nanocomposites have a size range from 30 to 60 nm with an average diameter of about 45 nm". The size distribution is quite broad. Is there any way to study the size effect of nanocomposites on the antibacterial or photocatalytic degradation efficacy? Any way to synthesis nanocomposites with more uniform size?

Author Response

REVIEWER 2

Comments and Suggestions for Authors

The manuscript presents an efficient hydrothermal method for the synthesis of BiPO4/ZnO nano-composites, which possess antibacterial effects and good photocatalytic performances for MB dye degradation. There are a few things need to be addressed:

QUERRY

  1. The format of some references (20, 23, 27, and so on) is incorrect. The family name must be written in full and initials used to represent given (first) names. Also need to revise the sections citing the reference in the manuscript (for example: XXX et al).

Author response: Authors are thankful to the reviewers to provide these valuable suggestions. The necessary corrections were carried out and incorporated into the revised manuscript.

QUERRY

  1. (page 4) The author mentioned Figure 3 before Figure 1 and 2 . The numbering of figures should follow the order which they were mentioned in the manuscript.

Author response: Following this reviewer’s suggestion we changed the numbers of figures in the revised version of the manuscript.

QUERRY

  1. (page 6) The author mentioned "The results demonstrate that as-prepared nanocomposites BiPO4/ZnO-NCs (B1Z4-75:300; NCs-4) caused maximum growth inhibition of 20 mm against the MDRs bacteria A. baumannii and a minimal of 12 mm against B. filamentosus at 80 μg mL−1 concentrations than the NPs and NCs displayed in Figure. 4." Any explanation of why NCs-4 is better than other NCs and NPs in antibacterial efficacy? Does NCs-4 show best antibacterial efficacy among all NCs and NPs at all concentrations and MDRs?

Author response: Several studies have reported the impact of Zinc oxide nanoparticles (ZnO-NPs) on bacterial pathogens.  Our research also clearly indicated that going to add ZnO-NPs ratios provides a better activity against MDRs compared to other ratios such as BiPO4/ZnO-NCs (B1Z1-300:300; NCs-1); BiPO4/ZnO-NCs (B2Z1-300:150; NCs-2); BiPO4/ZnO-NCs (B4Z1-300:75; NCs-3), BiPO4-NPs (NPs-1) and ZnO-NPs (NPs-2).

The different molar ratios of the Bimetallic nanocomposites (B1Z4-75:300; NCs-4) 1% of BiPO4 and 3% of ZnO-NCs shows the best antibacterial efficacy among all NCs and NPs at all concentrations with different (Gram ve+ and Gram ve) MDRs. 

QUERRY

  1. (page 6) In Table 1, the Probability > F* all larger than 0.05, the author should conclude explicitly that there is no sufficient evidence to say that there is a statistically significant difference in antibacterial activities among the four different concentrations. This is the purpose of ANOVA test.

Author response: Yes, we have accept the reviewer's suggestion. We also conclude that there is insufficient evidence to express statistically the significance of antibacterial activities at four different concentrations and mentioned it in the revised manuscript.

QUERRY

  1. (Page 7) The author mentioned "Fe-SEM images demonstrate that most of the particles are flower shape" but "SEM image of hexagonal and monoclinic BiPO4 and ZnO showed nanoparticles with mostly spherical and crystalline and polydispersed-like morphology", and "The obtained nanocomposite size distribution on BiPO4/ZnO using SEM also agrees well with the sizes provided by Fe-SEM“. Can the author explain why Fe-SEM has better ability than SEM to characterize the morphology of particles?

Author response: The difference between a Fe-SEM and an SEM lies in the electron generation system. As a source of electrons, the Fe-SEM uses a field emission gun that provides extremely focused high and low-energy electron beams, which greatly improves spatial resolution and enables work to be carried out at very low potentials. It has a better ability than SEM to characterize the morphology of particle size. The Fe-SEM gun that provides extremely focused high demonstrates that most of the nanocomposite particles are flower-shaped images. The preliminary SEM study confirmed the agglomeration of nanocomposites such as spherical and crystalline and polydisperse-like morphology. 

QUERRY

  1. In Figure 4, the author discussed the antibacterial efficacy of various NPs and NCs. Is there any SEM and Fe-SEM characterization of these NPs and NCs? Do they show nanoflower or other structures?

Author response: The main objective of the study was the synergistic effect of Multi-drug resistant strains (MDRs) through antimicrobial and enhanced photoexcited anti-microbial effects of BiPO4/ZnO. So the synthesized NPs and NCs particle size and distribution were characterized. This should be done via SEM and Fe-SEM in future studies.

QUERRY

  1. (page 7,13, Figure 6(c) and so on) The author mentioned "XX mints". If that's XX minutes, the abbreviation should be "XX mins".

Author response: This is corrected to “mints” in the revised version. The corrected figure was incorporated in the revised version of the manuscript.

QUERRY

  1. (page 7) The author mentioned "the prepared nanocomposites have a size range from 30 to 60 nm with an average diameter of about 45 nm". The size distribution is quite broad. Is there any way to study the size effect of nanocomposites on the antibacterial or photocatalytic degradation efficacy? Any way to synthesis nanocomposites with more uniform size?

Author response: The particle size distribution can be expressed in many ways based on different criteria, including number, diameter, area, volume, and mass. In our revised version of the MS we characterized the particle size distribution is characterized as a percentage of the number of particles in a specific size range compared to the total number of particles by measured Image J Software. 

Round 2

Reviewer 1 Report

The work is interesting, but the corrections made by the authors do not meet my expectations. First of all, the authors did not carry out an experiment that would allow to compare the activity of the catalyst with a standad catalytic system (TiO2-P25). In addition, the authors did not justify the purpose of determining the antimicrobial activity of the catalyst without irradiation.

Author Response

REVIEWER 1

Comments and Suggestions for Authors

Manuscript entitled "Synthesis of Bimetallic BiPO4/ZnO Nanocomposite: Enhanced Photocatalytic Dye Degradation and Antibacterial Applications" contains very interesting results. However, in my opinion, the manuscript in its current form should not be published. It requires significant corrections.

QUERRY

In order to assess the activity of the photocatalyst, the results should be compared with a known reference system. I suggest performing an analogous photocatalytic experiment in the presence of AEROXIDE® TiO2 P-25 (EVONIC, Germany; https://www.productcenter.coating-additives.com/pdf/regulatory/rds_ftp/AEROXIDE_TIO2_P_25.pdf). In addition, the text needs to be thoroughly redrafted. Large parts of it are chaotic (e.g. introduction) and/or unclear.

Author response: We rewrote several parts of the text, with particular focus on the INTRODUCTION.

Moreover, in order to assess the activity of the photocatalyst results, they were compared from ZnO, TiO2, Fe2O3, BiVO4, nano-heterojunction AgFeO2/ZnO, ZnO-based heterogeneous, Ag/ZnO heterostructural, polyaniline/CdO nanocomposite, Bi2WO6/BiVO4, Bi2WO6/BiOI p-n heterojunction have been widely used for the degradation of organic dyes and controlling microbes [37-46]. Compared to the standard of AEROXIDE® TiO2 P-25, our resultant nanopowders are also waterborne and have good solubility in water and white solid form only as well as the free-flowing powder with a 4.5 pH value. In addition, similar to AEROXIDE® TiO2 P-25, our resultant bimetallic BiPO4/ZnO nanocomposites have a very high specific surface area. Due to its unique ratio of flower crystalline structure, it is suitable for many catalytic and especially photocatalytic applications. This result clearly states that the produced nanostructures have equal potential with the standard AEROXIDE® TiO2 P-25 which will be applicable to photocatalytic degradation, especially on the organic pollutants and as it effectively disintegrates the toxicant that pollutes environmental waters such as surface and groundwater, causing significant pollution. The results demonstrated the BiPO4/ZnO-NCs (B1Z4-75:300; NCs-4) catalyst showed significantly superior photocatalytic activity under UV-light sources.

QUERRY

Abbreviations should not be used in the abstract.

Author response: We eradicated ABBREVIATIONS from the ABSTRACT.

QUERRY

No clear explanations for Table 1.

Author response: We provided clear explanations throughout the tables and also for Table 1. The level of inhibition should be expressed in relative units, e.g. %. We replaced inhibition levels by the relative unit “percentage”.

QUERRY

What is "mints" (L. 213, 214 etc.)?

Author response: This typo was replaced by mints, meaning minutes.

QUERRY

Chapter 3.3.3 lacks precise information on the preparation of the BiPO4/ZnO catalytic system.

Author response: We provided precise information of how the BiPO4/ZnO catalytic system was prepared. Catalyst designations in the text are convoluted and inconsistent. We arranged catalyst designations of the revised version in a consistent way.

QUERRY

Conclusions should not be a compilation of numerical results.

Author response: We revised the CONCLUSION substantially and avoided the mentioning of numerical results.

QUERRY

Where were the results described that were obtained using equation (4)?

Author response: The results described using equation (4) are shown in Table 2: Photocatalytic inactivation test by using bimetallic nanocomposites BiPO4/ZnO-NCs (B1Z4-75:300; NCs-4) catalyst under UV light exposure.

QUERRY

What was the purpose of the experiment using photocatalysts without irradiation (Chapters 2.2 and 3.5)?

Author response: The main purpose of the study is, to receive the adsorption/desorption equilibrium at the dark condition only in presence of the catalyst so that we performed the experiment, and after that only the suspension was irradiated by UV. A brief explanation of this procedure was given below:

Bimetallic nanocomposites BiPO4/ZnO-NCs (B1Z4-75:300; NCs-4) catalyst and multi-drug resistant strains (MDRs) suspension were mixed prior to light irradiation. Before UV light irradiation, the suspension (dye + bacterial strains) was magnetically stirred at 150 rpm constantly for 20 - 30 mints at dark conditions to obtain an adsorption/desorption equilibrium in the presence of the catalyst. After that, the suspensions were irradiated by UV light [Philips-UV light (150 W); λ=365 nm]. 

Round 3

Reviewer 1 Report

I propose to change the title to "Synthesis of Bimetallic BiPO4/ZnO Nanocomposite: Enhanced Photocatalytic Dye Degradation and Antibacterial Properties"

Author Response

REVIEWER 1

Comments and Suggestions for Authors

Manuscript entitled "Synthesis of Bimetallic BiPO4/ZnO Nanocomposite: Enhanced Photocatalytic Dye Degradation and Antibacterial Applications" contains very interesting results. However, in my opinion, the manuscript in its current form should not be published. It requires significant corrections.

QUERRY

In order to assess the activity of the photocatalyst, the results should be compared with a known reference system. I suggest performing an analogous photocatalytic experiment in the presence of AEROXIDE® TiO2 P-25 (EVONIC, Germany; https://www.productcenter.coating-additives.com/pdf/regulatory/rds_ftp/AEROXIDE_TIO2_P_25.pdf). In addition, the text needs to be thoroughly redrafted. Large parts of it are chaotic (e.g. introduction) and/or unclear.

Author response: We rewrote several parts of the text, with particular focus on the INTRODUCTION.

Moreover, in order to assess the activity of the photocatalyst results, they were compared from ZnO, TiO2, Fe2O3, BiVO4, nano-heterojunction AgFeO2/ZnO, ZnO-based heterogeneous, Ag/ZnO heterostructural, polyaniline/CdO nanocomposite, Bi2WO6/BiVO4, Bi2WO6/BiOI p-n heterojunction have been widely used for the degradation of organic dyes and controlling microbes [37-46]. Compared to the standard of AEROXIDE® TiO2 P-25, our resultant nanopowders are also waterborne and have good solubility in water and white solid form only as well as the free-flowing powder with a 4.5 pH value. In addition, similar to AEROXIDE® TiO2 P-25, our resultant bimetallic BiPO4/ZnO nanocomposites have a very high specific surface area. Due to its unique ratio of flower crystalline structure, it is suitable for many catalytic and especially photocatalytic applications. This result clearly states that the produced nanostructures have equal potential with the standard AEROXIDE® TiO2 P-25 which will be applicable to photocatalytic degradation, especially on the organic pollutants and as it effectively disintegrates the toxicant that pollutes environmental waters such as surface and groundwater, causing significant pollution. The results demonstrated the BiPO4/ZnO-NCs (B1Z4-75:300; NCs-4) catalyst showed significantly superior photocatalytic activity under UV-light sources.

QUERRY

Abbreviations should not be used in the abstract.

Author response: We eradicated ABBREVIATIONS from the ABSTRACT.

QUERRY

No clear explanations for Table 1.

Author response: We provided clear explanations throughout the tables and also for Table 1. The level of inhibition should be expressed in relative units, e.g. %. We replaced inhibition levels by the relative unit “percentage”.

QUERRY

What is "mints" (L. 213, 214 etc.)?

Author response: This typo was replaced by mints, meaning minutes.

QUERRY

Chapter 3.3.3 lacks precise information on the preparation of the BiPO4/ZnO catalytic system.

Author response: We provided precise information of how the BiPO4/ZnO catalytic system was prepared. Catalyst designations in the text are convoluted and inconsistent. We arranged catalyst designations of the revised version in a consistent way.

QUERRY

Conclusions should not be a compilation of numerical results.

Author response: We revised the CONCLUSION substantially and avoided the mentioning of numerical results.

QUERRY

Where were the results described that were obtained using equation (4)?

Author response: The results described using equation (4) are shown in Table 2: Photocatalytic inactivation test by using bimetallic nanocomposites BiPO4/ZnO-NCs (B1Z4-75:300; NCs-4) catalyst under UV light exposure.

QUERRY

What was the purpose of the experiment using photocatalysts without irradiation (Chapters 2.2 and 3.5)?

Author response: The main purpose of the study is, to receive the adsorption/desorption equilibrium at the dark condition only in presence of the catalyst so that we performed the experiment, and after that only the suspension was irradiated by UV. A brief explanation of this procedure was given below:

Bimetallic nanocomposites BiPO4/ZnO-NCs (B1Z4-75:300; NCs-4) catalyst and multi-drug resistant strains (MDRs) suspension were mixed prior to light irradiation. Before UV light irradiation, the suspension (dye + bacterial strains) was magnetically stirred at 150 rpm constantly for 20 - 30 mints at dark conditions to obtain an adsorption/desorption equilibrium in the presence of the catalyst. After that, the suspensions were irradiated by UV light [Philips-UV light (150 W); λ=365 nm]. 

QUERRY

I propose to change the title to "Synthesis of Bimetallic BiPO4/ZnO Nanocomposite: Enhanced Photocatalytic Dye Degradation and Antibacterial Properties"

Author response: Agree to change the title.
